Genome-wide identification and expression analysis of the OSC gene family in Platycodon grandiflorus

Wang Xiaoqin 1 2
Yan Dong 3
Chen Ling 762916349@qq.com 1 2
1 Key Laboratory of Exploitation and Utilization of Traditional Chinese Medicine Resources of Mianyang , Mianyang , Sichuan , China
2 School of Pharmacy, Sichuan College of Traditional Chinese Medicine , Mianyang , Sichuan , China
3 Department of Basic Medical, Sichuan College of Traditional Chinese Medicine , Mianyang , Sichuan , China
Nunes-da-Fonseca Rodrigo
Electronic publication date: 2024 Oct 28
Publication date: 2024
Volume: 12
Electronic Location ID: e18322
Received 2024 May 16; Accepted 2024 Sep 24
Copyright: ©2024 Wang et al.
Copyright year: 2024
Copyright holder: Wang et al.
License: This is an open access article distributed under the terms of the Creative Commons Attribution License, which permits unrestricted use, distribution, reproduction and adaptation in any medium and for any purpose provided that it is properly attributed. For attribution, the original author(s), title, publication source (PeerJ) and either DOI or URL of the article must be cited.
License URL: https://creativecommons.org/licenses/by/4.0/

Keywords: Triterpenoids, Oxido Squalene Cyclase (OSC), Genome-wide analysis, Platycodon grandiflorum

Funding: The Teacher Research Projects of Sichuan College of Traditional Chinese Medicine 22ZRYB02 The Project of Sichuan Traditional Chinese Medicine Culture Inheritance and Research Center 2021Z002 The Sichuan College of Traditional Chinese Medicine Doctoral Scientific Research Foundation 22BSZR02 Mianyang Key Laboratory of Traditional Chinese Medicine Resource Development and Utilization Open Project Fund 24ZYKF08 This work is supported by the Teacher Research Projects of Sichuan College of Traditional Chinese Medicine (Grant No. 22ZRYB02), the Project of Sichuan Traditional Chinese Medicine Culture Inheritance and Research Center (Grant No. 2021Z002), the Sichuan College of Traditional Chinese Medicine Doctoral Scientific Research Foundation (Grant No. 22BSZR02), and the Mianyang Key Laboratory of Traditional Chinese Medicine Resource Development and Utilization Open Project Fund (Grant No. 24ZYKF08). The funders had no role in study design, data collection and analysis, decision to publish, or preparation of the manuscript.

==============================
Platycodon grandiflorus stands as one of the most extensively utilized traditional Chinese medicinal herbs, with triterpenoids and their derivatives serving as its primary medicinal components. Oxido squalene cyclase (OSC), serving as a crucial enzyme in the triterpenoid synthesis pathway, has the capability to enzymatically generate significant quantities of sterols and triterpenoid intermediates. While the OSC gene family has been identified in numerous species, bioinformatics research on this family remains scant. Presently, the specific members of this gene family in Platycodon grandiflorus have yet to be definitively determined. In this study, we successfully identified a total of 15 PgOSC genes within the genome of Platycodon grandiflorus by conducting homology comparisons. These genes were discovered to be unevenly distributed across the five chromosomes of the species, organized in the form of gene clusters. Subsequently, we conducted a thorough analysis of the OSC gene family’s evolutionary relationship by constructing a phylogenetic tree. Other characteristics of PgOSC family members, including gene structure, conserved motifs, protein three-dimensional structure, subcellular localization, and cis-acting elements were thoroughly characterized. Furthermore, We analyzed the expression of PgOSC gene in different tissues of Platycodon grandiflorus by qRT-PCR, and found that the expression of PgOSC genes in root was higher than that in stem and leaf. Upon comparing the effects of salt, heat, and drought treatments, we observed a significant induction of PgOSC gene expression in Platycodon grandiflorus specifically under salt stress conditions. In summary, this study comprehensively identified and analyzed the OSC gene family, aiming to provide basic biological information for exploring the members of PgOSC gene family.

Introduction

Triterpenes, renowned as active components in numerous traditional Chinese medicinal plants, are a group of natural compounds with varied structures and biological activities including anti-tumor, anti-inflammatory, and immune-modulating properties (Wang et al., 2022).The formation of the triterpenoid carbon skeleton serves as a crucial step in generating the diverse array of triterpenoid products. Oxido squalene cyclase (OSC), serving as a crucial enzyme in the triterpenoid synthesis pathway, has the capability to enzymatically generate significant quantities of sterols and triterpenoid intermediates starting from 2,3-squalene oxide via a sequence of protonation, cyclization, rearrangement, and deprotonation steps (Abe, 2007; Haralampidis, Trojanowska & Osbourn, 2002; Shibuya et al., 1999). This catalytic activity is regarded as one of the most intricate biological processes with extensive potential for applications in the biomedical field. The specific conformation of OSC dictates its diverse catalytic orientations. In sterol biosynthesis, protosteryl cations are cyclized from 2,3-oxidosqualene through the “chair–boat–chair” conformation (CBC) of OSC to cycloartenol, lanosterol, and cucurbitadienol, etc. In triterpene biosynthesis, the dammarenyl cation is cyclized from 2,3-oxidosqualene through the “chair–chair–chair” conformation (CCC) of OSC to lupeol, α-amyrin, β-amyrin, and friedelin, etc (Parveen et al., 2024).

Plants abound in a diverse array of triterpenoids, ranking as one of the most prevalent secondary metabolites and commonly manifesting in the form of glycosides or esters. Portedpenes (Christianson, 2017; Chen et al., 2011). The triterpenoid skeletons in plants primarily encompass cucurbitanes, damaranes, lupines, ursanes, and friedelanes, based on their structural characteristics (Hill & Connolly, 2020). Triterpenoids have been proven to be the main active compounds in numerous medicinal plants, and they are currently being widely employed as therapeutic agents, due to their diverse biological activities. The activities of triterpenoids include significant anti-inflammatory effects (Alfaifi et al., 2020; Zhao et al., 2021), anticancer properties (Lombrea et al., 2021) suppressive effects on diabetes (Shih et al., 2013) cardio-protective functions (Camer et al., 2016), hepato-protective benefits (Xu et al., 2018; Hsiang et al., 2015), antibacterial capabilities (Akbar & Malik, 2002), antiviral activities (Xiao et al., 2018), and antiparasitic abilities (Rocha e Silva et al., 2015; Isah et al., 2016). As a result of these advantages, triterpenes hold immense potential for application in the field of medicine.

Furthermore, triterpenoids play a pivotal role in promoting plant growth and enhancing the plants’ resilience to stressful environment. For instance, specialized triterpenoids present in Arabidopsis possess the capability to regulate the growth of its roots, unaffected by hormonal signals (Bai et al., 2021). In response to waterlogging stress, the newly differentiated aerenchymatous phellem of soybean (Glycine max) accumulated a substantial amount of terpenoids (lupeol and betulinic acid) which contribute to effective internal aeration and root development for adaptation to waterlogged conditions. The accumulation of triterpenoids within the epidermal cuticle of aerial organs of Artemisia annua serves as a crucial factor in enhancing its resilience to stressful environments (Moses et al., 2015). In addition, triterpenoids also play a pivotal role in bolstering plant resistance to biological stress (Kuzina et al., 2009).

Platycodon grandiflorus (Jacq.) A. DC. (P. grandiflorus), belonging to the Campanulaceae family, stands as one of the most widely used traditional Chinese medicines, attributed to its diverse pharmacological effects including the promotion of lung function, resolution of phlegm, facilitation of throat comfort, and expulsion of pus (Ji et al., 2020). The triterpenoid saponins in Platycodon grandiflorus, also known as platycodon saponins, are the main medicinal chemical components of Platycodon grandiflorus, of which platycodon saponin D is the most important marker compound for quality assessment of platycodon grandiflorus. In this study, we performed a comprehensive genome-wide identification of the OSC gene family in the Platycodon grandiflorus, resulting in the discovery of 15 candidate OSC genes. Subsequently, we conducted detailed studies encompassing various aspects such as motifs, gene structure, phylogenetic relationships, and cis-elements, etc. Quantitative reverse-transcription PCR (qRT-PCR) analysis was employed to further validate the expression patterns of PgOSC genes across various organs and under different abiotic stress conditions. This investigation provides a robust foundation for future studies aimed at the catalytic mechanisms of OSC and advance the exploration of the medicinal value of Platycodon grandifloras.

Materials and Methods

Identification of OSC genes in Platycodon grandiflorus

The genome sequences of Platycodon grandiflorus downloaded from China National center for Bioinformation (https://www.cncb.ac.cn/), Accession number: GWHARYT00000000. The published protein sequence of OSC family members of Arabidopsis thaliana, Panax ginseng, Taraxacum mongolicum were downloaded from the National Center for Biotechnology Information database (http://www.ncbi.nlm.nih.gov/). Subsequently, OSC protein sequences from different species (Arabidopsis thaliana, Panax ginseng, Taraxacum mongolicum) were used as templates for sequence alignment in the genome of Platycodon grandiflorus, candidate OSC family genes were generated from the intersection of sequence alignment by ToolKit Biologists Tools (TBtools) software (https://github.com/CJ-Chen/TBtools). The identified PgOSC gene sequence information were listed in Table S2. The molecular formula, molecular weight, and isoelectric point were from Expasy (https://web.expasy.org/protparam/).

Phylogenetic analysis and classification of PgOSC genes in Platycodon grandiflorus

The OSC family protein sequences of A. thaliana (six), S. barbata (five), Platycodon grandifloras (15), T. coreanum (five), Panax ginseng (12) were downloaded from National Center for Biotechnology Information (https://www.ncbi.nlm.nih.gov/) and listed in Table S3. The multiple amino acid sequence alignment of OSC proteins was performed, and the phylogenetic tree was constructed by neighbor joining (NJ) method using ClustalW method implemented in Molecular Evolutionary Genetic Analysis software (MEGA 7). The generated tree was displayed using the iTOL website (https://itol.embl.de/upload.cgi).

Collinearity analysis and chromosomal mapping of the PgOSC family genes

The annotation information and the whole genome protein sequences of A. thaliana, Panax ginseng and T. mongolicum obtained from National Center for Biotechnology Information database (http://www.ncbi.nlm.nih.gov/). The collinearity relationships of the OSC genes between Platycodon grandiflorus and Rabidopsis thaliana, T. mongolicum and Panax ginseng genomes were determined and visualized by using the Multiple Collinearity Scan toolkit (MCScanX) of TBtools software with default parameters. The physical positions of PgOSC genes on the chromosomes of the Platycodon grandiflorus genome were determined and visualized using the TBtools software according to the genome annotation file.

Gene structure, motif analysis and cis-elements analysis

The whole genome sequences and CDS sequences of PgOSC genes were downloaded and subsequently utilized for in-depth gene structure analysis using the TBtools software. The motif analysis of the PgOSC family proteins was conducted utilizing the MEME website (http://meme-suite.org/tools/meme), while the domain analysis of the same protein family was performed through the Batch CD-Search tool available at (https://www.ncbi.nlm.nih.gov/Structure/bwrpsb/bwrpsb.cgi). The promoter sequences, encompassing 2000 base pairs upstream of the ATG start codon, were extracted from the PgOSC genes and subsequently analyzed for cis-elements using the Plant CARE platform (http://bioinformatics.psb.ugent.be/webtools/plantcare/html/). The types and quantities of these cis-elements were listed in Table S4.

Protein three-dimensional structure prediction of PgOSC proteins

The three-dimensional structures of PgOSC proteins were precisely modeled utilizing the SWISS-MODEL interactive workspace software (https://swissmodel.expasy.org/interactive). To ensure the accuracy and rationality of the homologous modeling outcomes, the SAVESv6.0 platform (https://saves.mbi.ucla.edu/) was employed for a comprehensive assessment of the modeling results.

Plant treatment and quantitative RT-PCR

One-year-old Platycodon grandiflorus plants were utilized to assess the expression levels of the PgOSC genes across various tissues, encompassing the stem, leaf, root, and beard root. To investigate the expression patterns of the PgOSC genes under various stress conditions, one-year-old potted Platycodon grandiflorus plants were subjected to salt stress (100 mM NaCl), heat stress (37 °C for 2 h), and drought stress (2 weeks). Subsequently, the expression level of the PgOSC genes were analyzed to assess its response to these stress environments. Total RNA was extracted from Platycodon grandiflorus roots using RNAiso Plus (Takara, Kusatsu, Japan). The extracted RNA was ultimately dissolved in 30µL of DEPC-treated water. Subsequently, 1.5 µg of total RNA was utilized for cDNA synthesis, and dilutions of this cDNA were employed in real-time RT-PCR experiments. The expression levels of the tested genes were normalized to PgACTIN (a constitutively expressed gene). The gene expression level was calculated by the formula (Econtrol)controlCT/(ETarget)TargetCT (Livak & Schmittgen, 2001).

Results

Identification of the OSC Gene Family in the Platycodon grandiflorus Genome

To detect potential oxidized squalene cyclases (OSC) in Platycodon grandiflorus, we utilized two OSC protein sequences of Arabidopsis thaliana (AT5G48010, AT4G15370) as reference templates and aligned them against the protein database of Platycodon grandiflorus. As a result, we identified 17 promising candidate proteins for OSC in Platycodon grandiflorus, and consistent results were achieved when employing the OSC protein sequence of Panax ginseng or T. mongolicum as an alternative template for alignment, and two members with few amino acids were eliminated by subsequent gene structure and motif analysis. Consequently, we successfully identified 15 OSC members in Platycodon grandiflorus and provided an overview of their physicochemical properties in Table 1.

Table 1 PgOSC gene family gene and protein properties.

Gene name	Gene ID	Chr	Number of amino acids	Mw (KD)	pI	Instability index	Aliphatic index	GRAVY	
PgOSC1	PGRA_12121	1	762	87301.84	6.56	36.87	82.05	−0.352	
PgOSC2	PGRA_10657	1	762	87492.03	6	41.3	79.23	−0.376	
PgOSC3	PGRA_10179	1	691	79542.33	6	42.02	80.88	−0.282	
PgOSC4	PGRA_11512	1	741	85015.2	6.03	38.01	77.54	−0.33	
PgOSC5	PGRA_21798	2	765	87104.19	6.46	40.93	82.14	−0.275	
PgOSC6	PGRA_08913	4	756	86093.26	5.9	45.43	82.17	−0.344	
PgOSC7	PGRA_09445	4	766	88087.45	5.85	48.83	75.67	−0.383	
PgOSC8	PGRA_07489	4	847	96453.28	6.15	46.9	74.17	−0.376	
PgOSC9	PGRA_07650	4	724	83650.59	6.48	44.71	70.75	−0.445	
PgOSC10	PGRA_09213	4	762	88002.6	5.94	48.12	75.42	−0.373	
PgOSC11	PGRA_14049	7	733	83987.98	6.13	33.97	88.34	−0.248	
PgOSC12	PGRA_00189	8	757	86921.1	6.08	45.86	82.5	−0.319	
PgOSC13	PGRA_00628	8	756	86721.19	6.29	46.21	77.18	−0.34	
PgOSC14	PGRA_00290	8	757	86147.42	6.26	41.15	83.36	−0.294	
PgOSC15	PGRA_01400	8	611	69135.51	7.84	43.78	87	−0.128	

Despite a total of nine chromosomes in Platycodon grandiflorus, the PgOSC members are unevenly distributed among just five of them (Chr 1, Chr2, Chr 4, Chr 7, Chr 8). The lengths of the 15 identified PgOSC proteins to range from 611 to 847 aa, and their molecular weights range from 69.14 kDa (PgOSC15) to 96.45 kDa (PgOSC8). The isoelectric point ranged from 5.85 (PgOSC7) to 7.84 (PgOSC15), and the instability index ranged from 33.97 (PgOSC11) to 48.83 (PgOSC7), and the Aliphatic index ranged from 70.75 (PgOSC9) to 88.34 (PgOSC11). In addition, the hydrophilicity values of all PgOSC proteins are less than 0, indicating that all of them are hydrophilic proteins, and amino acid hydrophilicity of all PgOSC proteins were listed in Table S1.

Chromosomal location and collinearity analysis of OSC family genes in Platycodon grandiflorus

The distribution of PgOSC genes on nine chromosomes in Platycodon grandiflorus was predicted based on gene assembly information published previously. Not all chromosomes of Platycodon grandiflorus, PgOSC family genes are specifically distributed across only five chromosomes, including four OSC genes on chromosomes 2 and 8, five OSC genes on chromosome 4, and only one OSC gene each on chromosomes 2 and 7 (Fig. 1). Furthermore, PgOSC genes are predominantly distributed in the form of gene clusters on the chromosomes of Platycodon grandiflorus. The gene clusters on chromosomes 4 and 8 contain up to four paralogous OSC genes, whereas the two gene clusters situated on chromosome 1 comprise only two members each. The clustering distribution of PgOSC family genes implies their functional similarity, indicating that they may share common biological roles and regulatory mechanisms.

Figure 1 The position of PgOSC genes on chromosomes of Platycodon grandiflorus.

The green bars signify chromosomes, with its corresponding name labeled above. The scale situated on the left side provides a measure of the genetic distance. The PgOSC gene family members are marked on chromosomes.

Phylogenetic relationship analysis of OSC proteins

To explore the phylogenetic relationships between the OSC proteins, a phylogenetic tree was constructed by MEGA7 software using OSC proteins sequence from A. thaliana (six), S. barbata (five), Platycodon grandifloras (15), T. coreanum (five), Panax ginseng (12), respectively. A total of 43 OSC protein sequences, originating from five diverse species, were categorized into four distinct subfamilies (Fig. 2). Four OSC proteins belonged to the subfamily I are all from A. thaliana, these may be related to the different evolutionary directions of A. thaliana and medicinal plants. Subfamily III contains the largest number of PgOSC proteins (six) and a small amount of proteins from other species. In addition, subfamily II encompasses 14 proteins, while subfamily IV comprises 16 proteins and the majority of which originate from Platycodon grandiflorus and Panax ginseng.

Figure 2 The phylogenetic analysis of OSC family genes in different species.

The NJ (Neighbor-Joining Method) tree was constructed from the protein sequences of OSCs using MEGA7. The subfamilies are marked with different colors.

Collinearity analysis of OSC family members

To elucidate the collinear relationship among OSC family gene species, we conducted a collinearity analysis employing the genomes of A. thaliana, Panax ginseng, S. miltiorrhiza, and Platycodon grandiflorus. Within this analysis, the OSC family members were emphasized through the utilization of red lines. As shown in Fig. 3, There is no collinearity between the OSC genes of Platycodon grandiflorus and A. thaliana, suggesting that they may not have a common ancestral gene. However, both Panax ginseng and S. miltiorrhiza, renowned for their medicinal properties, exhibit a distinct collinear relationship with Platycodon grandiflorus, such as PgOSC15 (PGRA_01400, Chr 2) in Platycodon grandiflorus and EVM0005942.1 (Chr 5) and EVM0039524.1 (Chr 12) in Panax ginseng; PgOSC9 (PGRA_07650 Chr 4) in Platycodon grandiflorus and EVM0056770.1 (Chr 3) and EVM0062132.1 (Chr 19)in Panax ginseng. These findings suggest that the OSC family genes of medicinal plants, including Platycodon grandiflorus, Panax ginseng, and S. miltiorrhiza, are more intimately related in the evolutionary process compared to those of A. thaliana. Unfortunately, hampered by the limitations of incomplete genomic data available in public databases, we were unable to carry out an intra-species collinearity analysis of Platycodon grandiflorus.

Figure 3 Interspecific collinearity relationship between PgOSC gene family members and A. thaliana, P. ginseng, and S. miltiorrhiza.

The collinear relationship between the OSC gene family members of different species and the PgOSC gene family members is connected by red lines.

Analyses of gene structures and conserved motifs of PgOSC genes

In order to reveal the structural characteristics of PgOSC family genes, we analyzed the exon–intron structures and the conserved motifs of 15 PgOSC genes using the MEME online tool and TBtools software. As shown in Fig. 4A, All 15 PgOSC proteins shared seven common motifs (motif 1 to motif 7), Motif 8 is present in all PgOSC gene family, with the exception of pgosc15. The exon-intron structures of PgOSC genes were analyzed by the CDS and genome sequences. As shown in Fig. 4B, The OSC family genes of Platycodon grandiflorus typically exhibit intricate gene structures, characterized by diverse combinations of numerous exons and introns. The length of PgOSC family genes surpasses 10 kb, excluding PgOSC2, PgOSC7, and PgOSC10. The extended intron structure accounts for the augmented gene length and implied that these genes may undergo more extensive differentiation and accumulate longer-term variations, thereby elevating the recombination frequency during OSC gene replication. This phenomenon is conducive to the evolution of organisms. However, given the intricate nature of the gene structure, it is imperative to take into account the quality of the database. This necessitates the corroboration of sequencing results from diverse sources to ensure the reliability of the data.

Figure 4 Conserved motifs and gene structure of PgOSC gene family members.

(A) Conserved motifs of PgOSC gene family members, the different colored boxes indicate different conserved motifs. (B) Gene structure of PgOSC gene family members, The green boxes represent UTRs, the blue boxes represent exons and the gray lines represent introns. The axes at the bottom are used to compare the lengths of different genes and proteins.

Prediction of the three-dimensional structures of PgOSC proteins

In order to further study the characteristics of tertiary structure of OSC protein in Platycodon grandiflorus, we conducted homology modeling analysis using the SWISS-MODE online tool. As shown in Fig. 5, The fundamental structure of OSC protein comprises a small β-fold block that is enclosed within two dense clusters of α-helices and the irregular curl shuttling through it seem to maintain the stability of the entire structure. Within the PgOSC family, significant variations in the number of central β-sheets exist among different members, potentially influencing the functional properties of these proteins. Furthermore, the stereoscopic structure of PgOSC12 and PgOSC13 distinctly differs from other members in Platycodon grandiflorus, consisting of a unique complex that integrates four fundamental PgOSC structures. Determining whether this distinctive architecture indeed bestows PgOSC12 and PgOSC13 with superior catalytic efficiency necessitates meticulous experimental validation.

Figure 5 Three dimensional structure model of PgOSC family member proteins.

Blue represents α-helix, green represents, β-fold, cyan represents β-corner, and white represents irregular curl. The structural model is generated through the SWISS-MODEL online website (https://swissmodel.expasy.org/interactive).

Prediction of subcellular localization of PgOSC family proteins

To obtain the predictive information of subcellular localization of PgOSC family proteins in Platycodon grandiflorus, we utilized the protein sequence of PgOSC to predict its subcellular localization using the WOLF PSORT website. The subcellular localization prediction of PgOSC family members are shown by heatmap. As shown in Fig. 6, Through clustering analysis, it becomes evident that PgOSC proteins exhibit three distinct patterns of subcellular distribution within cells. One pattern involves localization in mitochondria and peroxisomes, encompassing PgOSC1, PgOSC4, PgOSC13. Another pattern is found in the cytoplasm and nucleus, with high credibility in prediction results, such as PgOSC2, PgOSC5, PgOSC7 PgOSC11, PgOSC12, PgOSC15. Lastly, the remaining PgOSC family members are distributed across chloroplasts, vacuoles, Golgi apparatus, and other organelles.

Figure 6 Subcellular localization prediction of PgOSC family proteins.

PgOSC family protein sequences were predicted for subcellular localization using the WoLF PSORT website. The deepening of the red color indicates an increasing probability. Plas, plasma membrane; Cyto, cytoplasm; Chlo, chloroplast; Mito, mitochondrion; Pero, peroxisome; E.R., Endoplasmic Reticulum; Vacu, Vacuole; Nucl, nucleus.

Analysis of cis-acting elements of PgOSC promoter

Cis-elements are important for gene transcription, to acquire information of the Cis-elements on PgOSC gene promoters , a 2000 bp DNA sequence upstream of each gene translation start site extracted from the Platycodon grandiflorus genome database were used to make a prediction through the PlantCARE website (Fig. 7). A moderate level of adversity can promote the accumulation of secondary metabolites. We found that the PgOSC family contains a rich set of cis-acting elements that respond to drought, low temperature, and hypoxia, which are beneficial for the accumulation of triterpenoids. In addition, hormone response elements involved in auxins, abscisic acid, gibberellins, and jasmonic acid indicate that hormones remain important factors in regulating the growth and development of Platycodon grandiflorus and the accumulation of secondary metabolites. Details of cis-acting elements in the promoter region of PgOSC genes shown in Table S4.

Figure 7 Analysis of cis-acting elements in the PgOSC gene family promoter.

The distribution of cis-acting elements in the PgOSC promoter. Boxes of different colors and shapes represent different types of cis-acting elements. The scale at the bottom indicate the length of the promoter sequence (0∼2000 p) and the location of cis-acting elements.

Expression pattern analysis of PgOSC genes

To ascertain the tissue expression pattern of PgOSCs, we employed qRT-PCR to assess the expression levels of PgOSC family genes across leaf, steam, root, and root beard of Platycodon grandiflorus. As expected, the expression level of PgOSC family genes was higher in the roots of Platycodon grandiflorus than in its stem, leaf and root beard tissues. There was no significant difference in the expression of OSC family genes between stems and leaves (except PgOSC8) (Fig. 8) . These results suggest that The root of Platycodon grandiflorus may function as the primary site for triterpenoids synthesis, which aligns with the utilization of the root specifically for medicinal purposes, rather than other tissues of the plant.

Figure 8 Analysis of gene expression level of PgOSC genes in different tissues of Platycodon grandiflorus.

Data are the means ± standard error (n = 3). Asterisks indicate the P value determined by a Student’s t test as compared to control (leaf)(* P < 0.05, ** P < 0.01).

Moderate stress conditions can stimulate the accumulation of secondary metabolites in plants.To clarify the relationship between the synthesis of triterpenoids in Platycodon grandiflorus and stress conditions, we subjected one-year-old plants to various types of stress treatments, including heat, drought, and salt. Subsequently, we used qRT-PCR to detect changes in the expression of PgOSC genes. Excitingly, salt stress emerged as a prominent factor, exhibiting a robust induction effect on most genes of the PgOSC family (excluding PgOSC3, PgOSC8, PgOSC15) (Fig. 9). This result suggests that salt stress effectively promotes the accumulation of triterpenoids in Platycodon grandiflorus. In addition, drought stress specifically induced the expression of PgOSC14 and PgOSC15.

Figure 9 Analysis of gene expression level of PgOSC genes under different stress conditions (heat, drought and salt).

Data are the means ± standard error (n = 3). Asterisks indicate the P value determined by a Student’s t test as compared to control (* P < 0.05, ** P < 0.01).

Discussion

Triterpenoids are compounds that commonly occur in plants, either in their free state or combined with sugars in the form of glycosides or esters. Possessing a diverse array of biological activities, they serve as pivotal effective components in numerous Chinese medicinal materials. For instance, ginsenosides possess the ability to foster protein biosynthesis, modulate metabolic functions, strengthen immune responses, and effectively suppress tumor growth (Goodwin & Best, 2023). The multifaceted functionalities of saikosaponin encompass its potent anti-inflammatory properties, along with its capability to effectively reduce cholesterol and triglyceride concentrations in the plasma (Wang & Li, 2023). Aescin possesses anti-exudative, anti-inflammatory, and anti-congestion properties, thereby exhibiting significant therapeutic potential in addressing a range of physiological conditions (Savarino et al., 2023). Glycyrrhizin exhibits adrenocortical hormone-mimetic effects, and exhibits prophylactic potential against liver cirrhosis, atherosclerosis, and ulceration (Li et al., 2020).

OSC family genes encode oxidized squalene cyclase, a pivotal enzyme serving as a branching point in the biosynthetic pathways of sterols and a diverse array of saponins within organisms. OSC possesses the remarkable ability to catalyze the formation of sterol or triterpene skeletons, characterized by their distinct parent ring structures, and subsequently, through a cascade of enzymatic reactions, these skeletons give rise to a plethora of functionally diverse compounds. In A. thaliana, a total of 13 OSC genes have been identified, all capable of catalyzing the synthesis of triterpenoid products. These products include cycloartenol, lanosterol, β-amyrin, marneral, thalianol, and arabidiol, demonstrating the diverse biosynthetic capabilities of these enzymes within the plant species (Thimmappa et al., 2014). Heterologous expression of OSC genes in Oryza sativa leads to the synthesis of a range of products, including β-amyrin, parkeol, and isoarborinol, among others (Inagaki et al., 2011).

Employing OSC genes derived from A. thaliana, O. sativa, and Panax ginseng as comparative templates, we conducted a homology-based screening of the Platycodon grandiflorus genome, ultimately identifying 15 candidate genes belonging to the OSC family (Table 1).

Gene families typically consist of numerous copies of an ancestral gene that have arisen through replication or mutational events (Razin et al., 2021). Although the members of these gene families may occasionally cluster together, forming distinct gene clusters, they are more often distributed across various loci within the same chromosome or even reside on separate chromosomes (Shoji & Yuan, 2021). In this study, chromosomal localization analysis has demonstrated that the pgosc family genes are clustered on several chromosomes of Platycodon grandiflorus, specifically chromosomes 1, 4, 7, and 8 (Fig. 1). This clustering pattern may be attributed to the occurrence of unequal crossover events during meiotic recombination involving OSC ancestral genes. Such events presumably facilitate the coordinated expression of OSC family genes, thereby promoting their functional interactions and enhancing the overall functionality of this gene family.

Collinearity analysis between species reveals the homology and evolutionary relationship between different species, and predicts the similarity between genomes (Wang et al., 2012). In this study, we examined the genomes of Platycodon grandiflorus, Arabidopsis thaliana, Panax ginseng, and S. miltiorrhiza. Notably, we discovered that Platycodon grandiflorus exhibits a Collinearity relationship with Panax ginseng and S. miltiorrhiza, but not with A. thaliana (Fig. 2). Whether this observed pattern is linked to the selection pressure imposed by its utilization (A. thaliana cannot be used as medicine) remains a fascinating avenue for further exploration.

Inaccuracies in the pairing process, the insertion of exogenous DNA fragments, or the occurrence of gene recombination during DNA replication can give rise to the expansion of the intron region within a gene, resulting in a gradual enlargement of this non-coding segment within the genome (La Rosa et al., 2020; Mukhopadhyay, Wai & Hausner, 2023). Intron expansion events contribute significantly to the enhancement of genome complexity and diversity, offering organisms an expanded repertoire of genetic variations that facilitate their adaptation to diverse environments and survival challenges (Liu et al., 2021). The OSC family genes in Platycodon grandiflorus typically exhibit a distinct genomic architecture, characterized by the presence of lengthy introns interspersed with at least 15 exon regions (Fig. 4). While this structure is beneficial for the expression and evolution of PgOSC genes, it concurrently poses potential risks to the stability of gene function. Additionally, in dealing with complex genomic structures, it is necessary to cross-validate data from various sources to prevent one-sided conclusions that may arise due to limitations in sequencing technology.

Supplemental Information

Table S1 The prediction of amino acid hydrophobicity of PgOSC protein

Table S2 The list of proteins and CDS sequences of PgOSC

Table S3 The list of protein sequences for constructing phylogenetic tree

Table S4 The information of cis-acting elements on the promoter

Supplemental Information 5 MIQE Checklist

Additional Information and Declarations

Competing Interests

Author Contributions

Data Availability

The authors declare there are no competing interests.

Xiaoqin Wang conceived and designed the experiments, performed the experiments, analyzed the data, prepared figures and/or tables, authored or reviewed drafts of the article, and approved the final draft.

Dong Yan analyzed the data, prepared figures and/or tables, authored or reviewed drafts of the article, dong revised the manuscript in accordance with the editor’s comments and addressed the reviewer’s inquiries, and approved the final draft.

Ling Chen conceived and designed the experiments, authored or reviewed drafts of the article, and approved the final draft.

The following information was supplied regarding data availability:

All raw data and sequence information (DNA or protein) are available in the Supplemental Files.

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
