# Peer review of "Genome-wide identification and expression analysis of the OSC gene family in Platycodon grandiflorus"

_PeerJ, doi:10.7717/peerj.18322_

## Round 0.1 · original submission · Major Revisions

Dear Dr. Chen,
Please address reviewers comments and provide a detailed response to each question of both reviewers. Please focus on how the current manuscript is different from previous publications e.g. Oxidosqualene cyclase family has already been identified in P. grandiflorus by Kim et al. (2020;), among other observations provided.

Reviewer 1 ·

Basic reporting

Wang and Chen have identified and characterized 15 putative PgOSC, and examined their expression patterns. However, they provided only preliminary data (obtained by simple-database-mining) which is not sufficient to conclude that these putative genes are 'true OSC'.

Experimental design

At least, complementation analysis using Arabidopsis T-DNA mutants or enzyme characterization using fusion protein is needed.

Validity of the findings

the authors fail to show how their study uniquely contributes to the current body of knowledge; Oxidosqualene cyclase family has already been identified in P. grandiflorus by Kim et al. (2020; Whole-genome, transcriptome, and methylome analyses provide insights into the evolution of platycoside biosynthesis in Platycodon grandiflorus, a medicinal plant).

Additional comments

A reviewer believes that a submitted manuscript is a still premature state for the publication in “PeerJ”, especially in consideration of the journal’s impact factor noted in its own website.

Reviewer 2 ·

Basic reporting

no comment

Experimental design

no comment

Validity of the findings

no comment

Additional comments

The manuscript entitled " Genome wide identification and expression analysis of the OSC gene family in Platycodon grandiflorus" is comprehensive. The author has performed genome wide characterization of OSC genes in Platycodon grandiflorus. This study underlines the localization of OSC genes, their domain architecture as well as evolutionary relationship. However, I have certain suggestions which are enlisted below:
1. What is novel in this study as there are several reports available in OSC gene family? Does the novelty only involves different genotype with similar studies?
2. What are the novel findings in the species taken?
3. Why is this species taken for study and why chosen OSC gene family for study?
4. A total of 15 OSC gene family members were identified in the study, but only 10 were verified in the qRT-PCR experiment. I think the unverified 5 genes are also very important, and it is recommended to increase the experiment.
5. In the study of abiotic stress, the specific tissue sites were not specified.
6. What is the age of the plants treated with stress? If it is an annual plant, whether there is accumulation of triterpenoid saponins

---

## Round 0.2 · accepted · Accept

Dear Dr. Chen,

Congratulations on the acceptance of your manuscript.

Best regards

Reviewer 2 ·

Basic reporting

No comment

Experimental design

No comment

Validity of the findings

No comment

Additional comments

No comment